# Chloride Channels in Astrocytes: Structure, Roles in Brain Homeostasis and Implications in Disease

**DOI:** 10.3390/ijms20051034

**Published:** 2019-02-27

**Authors:** Xabier Elorza-Vidal, Héctor Gaitán-Peñas, Raúl Estévez

**Affiliations:** 1Unitat de Fisiologia, Departament de Ciències Fisiològiques, Genes Disease and Therapy Program IDIBELL-Institute of Neurosciences, Universitat de Barcelona, L’Hospitalet de Llobregat, 08907 Barcelona, Spain; xabier.ev@gmail.com (X.E.-V.); hektorgp@hotmail.com (H.G.-P.); 2Centro de Investigación en red de enfermedades raras (CIBERER), ISCIII, 08907 Barcelona, Spain

**Keywords:** astrocyte, chloride channel, human diseases, structure, physiology

## Abstract

Astrocytes are the most abundant cell type in the CNS (central nervous system). They exert multiple functions during development and in the adult CNS that are essential for brain homeostasis. Both cation and anion channel activities have been identified in astrocytes and it is believed that they play key roles in astrocyte function. Whereas the proteins and the physiological roles assigned to cation channels are becoming very clear, the study of astrocytic chloride channels is in its early stages. In recent years, we have moved from the identification of chloride channel activities present in astrocyte primary culture to the identification of the proteins involved in these activities, the determination of their 3D structure and attempts to gain insights about their physiological role. Here, we review the recent findings related to the main chloride channels identified in astrocytes: the voltage-dependent ClC-2, the calcium-activated bestrophin, the volume-activated VRAC (volume-regulated anion channel) and the stress-activated Maxi-Cl^−^. We discuss key aspects of channel biophysics and structure with a focus on their role in glial physiology and human disease.

## 1. Introduction

Chloride is the main physiological anion, serving as the principal compensatory ion for the movement of major cations such as Na^+^, K^+^ and Ca^2+^ [1,2]. Regarding the importance of Cl^−^ in the CNS, the functions attributed to chloride channels include the control of membrane potential, cell volume homeostasis and regulation of cell proliferation and apoptosis [3]. While extracellular Cl^−^ concentrations are fixed around 96–106 mM, intracellular Cl^−^ concentrations are more variable and depend on the transmembrane potential and the presence of secondary active Cl^−^ transporters. Thus, the presence and function of diverse chloride channels and transporters in different brain cell types will determine the direction of Cl^−^ fluxes and the subsequent cellular functions that Cl^−^ exerts. Whereas for neurons the impact of Cl^−^ currents on excitability is well understood [4], the role of Cl^−^ in astrocytes is less clear.

Astrocytes comprise the most abundant and diverse type of glial cell in the brain, being distributed throughout the grey and white matter. They are characterized by a highly complex morphology that allows the development of contacts with both neuronal soma and dendrites from neighbouring neurons as well as blood vessels via the astrocytic endfeet processes [5,6,7]. Astrocytes develop a variety of functions essential for brain homeostasis. Some of the functions assigned to astrocytes are the modulation of excitatory signals through gliotransmitters, the protection of neurons from oxidative stress, participation in stem-cell proliferation and axonal migration. Astrocytes also play a crucial role in the maintenance of the blood–brain barrier (BBB), and regulate the energetic metabolism coupling cerebral blood flow to neuronal needs and increasing the availability of oxygen and glucose [8,9,10]. They can also enhance the exchange of soluble substrates between the CSF and interstitial fluid in the brain in a process named glymphatic flow [11,12,13]. Moreover, astrocytes can control synaptic activity, as they can take up K^+^ and neurotransmitters such as glutamate during neuronal activity, facilitating fast repetitive neurotransmission, especially in Bergmann glia. Astrocytes have also been reported to be involved in regulating cell volume, pH and the buffering of extracellular K^+^ [6,14,15].

From among the different families of chloride channel activities with known proteins, here we review the main chloride channels found in astrocytes with a focus on their structure and their impact on astroglial physiology and disease. 

## 2. ClC-2

### 2.1. The Structure and Function of the ClC-2 Channel

Codified by the *CLCN2* gene, the ClC-2 channel belongs to the nine-member family of voltage-gated chloride channels and transporters (CLC) [16,17,18]. The oligomeric structure of ClC-2 is common to all CLC channels and transporters (Miller, 2003): they have a homodimeric double-barrelled structure, with each monomer contributing an identical pore/binding site. Each subunit consists of a membrane protein with 18 transmembrane helices followed by a cytosolic C-terminus containing two conserved cystathionine-β-synthase (CBS) domains [19] (Figure 1A). CBS domains are essential for correct intracellular trafficking and may be involved in the regulation of gating [20,21]. The structure of several CLC proteins has been solved, confirming the double-barrelled dimeric structure with independent pores, and revealing residues involved in chloride binding/coordination [19,22,23]. The permeability sequence of ClC-2 is Cl^−^ > Br^−^ > I^−^ > F^−^, similar to all CLC members [1,18,24].

ClC-2 is activated upon membrane hyperpolarization, presenting a slow time-course, inward rectifying currents, and its voltage-dependent gating can be modulated by Cl^−^ and H^+^. It can also be activated by hypotonic-induced cell swelling [18,25,26,27,28]. After activation at negative potentials, the current decreases at positive potentials, resulting in a slow current deactivation. Mutagenesis studies determined that the intracellular N-terminus of the channel is necessary for voltage- and pH-dependent activation [25]. Moreover, it has been reported that the activation and deactivation gating of ClC-2 is significantly accelerated by deletions or modifications of C-terminus domains, suggesting the modulatory function of this domain [29,30,31]. Two different mechanisms for channel gating have been proposed: (1) A fast gating process which occurs when a single protopore is activated independently due to a conformational change that can be controlled by voltage, Cl^−^ concentration and pH; and (2) a slow gating mechanism that opens both protopores (thus being called common gating) possibly through the cooperative movement of both CBS domains [24,32,33,34].

### 2.2. The Role of ClC-2 in Glial Physiology and Implications in Disease

ClC-2 is widely expressed in the organism, being found in the brain, kidney, pancreas, skeletal muscle and gastrointestinal tract among other tissues [18,31]. Given the ubiquitous expression of the ClC-2 channel, differential functions have been proposed depending on the cell type. In the brain, ClC-2 is expressed in pyramidal hippocampal neurons and interneurons [35], but also in astrocytes in the endfeet surrounding blood vessels [36,37], as well as in oligodendrocytes [38]. Here, ClC-2 is thought to act in physiological processes such as cell volume regulation, the control of intracellular Cl^−^ concentrations in inhibitory GABAergic neurons, and in the regulation of ionic homeostasis of Cl^−^ and K^+^ at astrocyte–oligodendrocyte junctions [1,27,35,38,39]. A direct role of ClC-2 in the control of neuronal excitability has been proposed, especially by preventing Cl^−^ accumulation at GABAergic synapses [40,41]. Despite its role in this physiological process, evidence that ClC-2 may be related to idiopathic generalized epilepsy remains controversial [42,43,44].

One of the hallmarks of *Clcn2* knockout mice is that the lack of ClC-2 induces the vacuolization of the external myelin sheaths [38] and alters hippocampal neurotransmission in ageing mice [45]. However, *Clcn2^−/−^* mice do not present major neurological defects [38]. The white matter vacuolization observed in these mice is strikingly similar to the vacuolizing phenotype of patients affected with Megalencephalic Leukoencephalopathy with subcortical Cysts (MLC) [46]. This prompted the study of a possible link between *CLCN2* mutations and MLC, but this hypothesis was discarded [47]. MLC disease is caused by mutations in either *MLC1* or *GLIALCAM* genes [48,49]. Interestingly, it was later reported that GlialCAM, the MLC-related glial adhesion molecule, is an auxiliary subunit of the ClC-2 channel [37]. In vitro, GlialCAM changes the sub-cellular localization of ClC-2, targeting the channel to cell–cell junctions. GlialCAM also modifies the ClC-2 electrophysiological properties, increasing total current amplitudes and abolishing the rectification of ClC-2, which is thus opened at positive voltages [37]. The fact that GlialCAM shows a more restrictive expression pattern than the ubiquitously expressed ClC-2 indicates that it may act as an auxiliary subunit specifically in glial cells. Interestingly, but probably without physiological relevance, GlialCAM is able to interact in vitro with the chloride channels ClC-1, ClC-Ka and ClC-0, opening the common gate and slowing its deactivation, but not with the chloride intracellular transporter ClC-5 [50]. 

Studies of *Glialcam^−^^/^^−^* mice revealed that ClC-2 protein levels are reduced in the cerebellum and that the channel is mislocalized in Bergmann glia (astrocytic processes surrounding blood vessels) and in oligodendrocytes [51,52]. Surprisingly, these defects are also present in the *Mlc1^−/−^* mouse model. Defects in ClC-2 localization in oligodendrocytes have also been described in *Mlc1^−/−^* mice, although MLC1 expression is restricted to astrocytes [52]. Furthermore, ClC-2 activity is inwardly rectifying in both *Glialcam^−/−^* and *Mlc1^−/−^* oligodendrocytes, revealing a functional relationship between MLC1/GlialCAM and ClC-2 in vivo in this cell type [52]. Conversely, ClC-2-mediated chloride currents recorded in astrocytic Bergmann glia are inwardly rectifying and the rectification remains inward in both *Glialcam^−/−^* and *Mlc1^−/−^* mice [52]. These results are rather surprising considering the clear ClC-2 protein mislocalization found in both animal models in Bergmann glia. It has recently been reported that in astrocytes, GlialCAM modifies ClC-2-mediated currents only under depolarizing conditions due to the formation of a ternary complex with MLC1 [53], suggesting temporary and controlled regulation of the formation of an astrocytic ClC-2/MLC1/GlialCAM ternary complex. This complex may be needed in certain depolarizing conditions, such as during high neuronal activity (Figure 1B) [53,54].

Recently, mutations in the *CLCN2* gene have been associated with a disease phenotype characterized by mild leukoencephalopathy, infertility and secondary paroxysmal kinesigenic dyskinesia [55,56,57,58,59]. This disease was recently renamed *CLCN2*-related leukoencephalopathy or CC2L [60]. *CLCN2* mutations related to CC2L have been reported to severely impair channel function and trafficking [61]. All the mentioned evidence regarding CC2L patients, the phenotype of *Clcn2^−/−^* mice, and the modulation of the ClC-2 channel through its subunit GlialCAM gives more support to the idea of a role of glial ClC-2 in controlling ionic homeostasis. In line with this idea, a similar vacuolizing phenotype caused by ClC-2 suppression is observed in the vacuolization caused by the ablation of astroglial connexins 32 and 47 or the lack of Kir4.1, prompting the hypothesis that ClC-2 may be involved in potassium syphoning [38,53]. This physiological mechanism disperses local high extracellular K^+^ concentrations through a glial syncytium, which is essential for the maintenance of extracellular K^+^ at a level compatible with continued neuronal function [62,63,64]. Astrocytic uptake is mediated by the inwardly rectifying K^+^ channel Kir4.1 [65,66], the co-transporter NKCC1 and mainly by the Na^+^/K^+^ ATPase pump [67,68,69]. Other essential elements in the process of K^+^ syphoning are gap junctions formed by connexins [70,71] and the water channels formed by AQP4 proteins [72,73]. This direct ionic entry is accompanied by an influx of anions such as Cl^−^, which preserves electroneutrality. This Cl^−^ entry may come from the co-transporter NKCC1 or through specific chloride channels. It has been suggested that spatial buffering is combined with approximately equimolar KCl transport [74]. Thus, it is in this accompanying Cl^−^ influx that the glial GlialCAM/ClC-2 channel may be required. The Kir4.1 potassium channel is the one that shows the least rectification within the different inward rectifier potassium channels, being able to mediate potassium flux in both directions. Similarly, the GlialCAM protein is needed to reduce the inward rectification of the chloride channel, permitting the efflux or influx of chloride according to electrochemical gradients. Therefore, a reduction in the degree of rectification of the ion channels involved in potassium syphoning may be a common functional feature needed to fulfil its role in this process. 

## 3. Bestrophin 1

### 3.1. Bestrophin 1 Structure and Function

Bestrophin1 (Best1) is a membrane protein that belongs to the family of calcium-activated chloride channels (CACC). Four homologues of this channel (*BEST1-4*) are found in humans [75]. This channel is encoded by the *VMD2* gene, which translates for a 585 amino acid protein. 

The three-dimensional channel structure was resolved for the prokaryotic Bacterial KpBEST and the eukaryotic Chicken BEST1, the latter crystallized in complexes with a Fab [76,77]. Common features of both structures include channels that assemble in pentameric complexes arranged symmetrically around an axis that forms a central pore. Each subunit has a complex topology, with a transmembrane domain comprised of four transmembrane helices (TM). A large cytoplasmic domain is formed by five alpha helices connecting TM2 and TM3. Finally, TM4 is connected to a C-terminal helix through a conserved carboxylate-rich loop [76,77] (Figure 2A). While the first part of the protein, comprising the transmembrane domain, is highly conserved between species, the intracellular C-terminus is more variable and may be involved in protein–protein interactions [78].

The pore of Best1 presents a complex configuration. It stretches from the extracellular domain, crosses the transmembrane region and continues through a large cytosolic domain of the pentamer. The diameter varies along its length, including two hydrophobic constrictions, named *neck* and *aperture,* which are thought to control the gating of the channel in response to calcium [76]. In fact, most disease-causing mutations in the Best1 protein are located in the amino acids that configure the neck [79]. The pentameric structure of Best1 also reveals a specific cytosolic region containing a cluster of acidic amino acids, which is a calcium-binding site. This site is named *Ca^2+^ clasp* and is configured by the N-terminal part of one subunit and the C-terminal portion of the adjacent subunit [76]. It has been determined that the function of the calcium clasp is to sense the concentration of intracellular calcium, while the neck of the pore acts as a gate for the calcium-dependent chloride permeation of the channel [80]. Like other chloride channels, Best1 allows the passage of several anionic molecules, with a sequence of relative anion permeability as it follows: SCN^−^ > NO_3_^−^ > I^−^ > Br^−^ > Cl^−^ [81].

### 3.2. Physiological Roles of Best1 in the Brain and Implications in Disease

Mutations in the *BEST1* gene were first reported in patients suffering from inherited forms of retinopathies, specifically in vitelliform macular dystrophy (BVMD), also named Best’s disease (reviewed in [82]). Hence, this channel is predominantly found in the retinal pigment epithelium (RPE) and most studies regarding Best1 have focused on its role in the retina [82].

Besides the RPE, mouse Best1 (mBest1) channel expression has been reported in the mouse brain, where it is specifically present in both neurons and astrocytes, and dorsal root ganglion [78,83,84]. In the cerebellum, mBest1 is found in Purkinje cells, Bergmann glia and lamellar astrocytes in the molecular layers [84,85]. This reported expression by means of Western Blot and immunohistochemistry contrast with the low levels of RNA detected in the transcriptomic analysis (https://web.stanford.edu/group/barres_lab/). However, recent studies using *best1* knockout mice as a control [86] corroborate the specificity of the observed expression of mBest1 in the cerebellum.

Interestingly, Best1 is expressed in astrocytic microdomains near neuronal synapses. At these locations, it has been reported that Best1 can release glutamate in a slow calcium- and GPCR-dependent manner [85,87], which then targets and activates hippocampal pyramidal neurons in order to modulate synaptic plasticity [85]. Besides glutamate, Best1 is also permeable to other neurotransmitters, including GABA [85,86]. Specifically, the tonic release of GABA from glial cells (astrocytes) through Best1 channels in the cerebellum exerts tonic neuronal inhibition (Figure 2B) [84]. This inhibition is important for neuronal information processing and has been associated with several pathologies. It has been suggested that this GABAergic release from astrocytes through Best1 is important for motor coordination in the cerebellum, and may modulate neuronal inhibitory/excitatory balance [84,86].

This GABA release from astrocytes through Best1 may be important in attempts to explain some pathological conditions where GABAergic signalling is altered. Astrocytic GABA is reported to be elevated in some conditions where reactive astrocytes are active, such as in Alzheimer’s disease [88]. Furthermore, there is a redistribution of mBest1 expression in those reactive astrocytes from a mouse model of Alzheimer’s [88], suggesting that GABA release through Best1 and the changes in its distribution may be relevant in the pathological mechanisms underlying these GABAergic changes in neurological diseases [88,89]. Regarding these dynamic changes in Best1, it has been found that 14-3-3γ protein binds directly to Best1 in astrocytes, regulating its surface expression [90]. The compound 14-3-3γ belongs to a family of regulatory molecules that promote plasma membrane targeting of several other receptors, and have previously been linked with the pathophysiology of some neurological disorders [91,92].

The finding of the permeability of large osmolytes such as GABA and glutamate through Best1 is in apparent contradiction to the current knowledge of the channel structure because the analyses of the ionic pore of the channel indicate that there is insufficient space for such large molecules (as discussed in [82]). It is possible that the explanation lies in the fact that the obtained structure represents the closed configuration of the channel. In fact, the channel permeability sequence, with higher permeability to large anions (SCN^−^, NO_3_^−^) than monovalent ions (Br^−^, Cl^−^) [93], already indicates that the relatively small pore found in the crystallized structures (which appears to be more permeable to smaller anions) may not be the only configuration of the channel. Recent work on the structural basis of the gating of bestrophin 1 [94] has shed some light on this issue. New cryo-EM structures of the chicken Best1 channel with different gating states of the channel have been achieved. These structures reveal a rearrangement of several amino acids of pore-lining helices that greatly enhance the neck of the pore (Figure 2C). It has been reported that this open configuration makes it plausible that Best1 may conduct GABA and other larger molecules [94].

## 4. The Volume-Regulated Anion-Channel (VRAC)

### 4.1. General Features

The volume-regulated anion channel (VRAC) is a ubiquitously expressed outwardly-rectifying anion and osmolyte channel crucial for cell volume homeostasis. Animal cells, especially in the presence of aquaporins, are relatively permeable to water. Osmotic gradients established between the intracellular fluid (ICF) and the extracellular fluid (ECF) passively drive water fluxes across membranes, thereby altering cell volume. In the absence of regulation, uncontrolled cell shrinkage or swelling could harm cells, from simply altering the concentrations of biochemical partners needed for proper cell function, to severe alterations in the cell’s integrity. Furthermore, cell volume changes are actively promoted by cells and constitute an integral part of well-controlled processes like the cell cycle, proliferation, migration, apoptotic volume decrease, trans-epithelial transport, etc. [95]. In order to control cell volume, the coordinated activity of a plethora of ion channels and transporters alters the osmotic gradient by the net influx or efflux of osmolytes through the processes of regulatory volume increase (RVI) or regulatory volume decrease (RVD) [96]. The VRAC is activated under cell swelling and/or signalling cascades and plays a pivotal role in the RVD process. Through the efflux of anions like Cl^−^, and organic osmolytes like glutamate, taurine, and possibly even ATP [97,98], the VRAC plays a critical role in sustaining the driving force for the complementary efflux of K^+^ by independent K^+^ channels. 

After decades of research, the molecular entities of the VRAC were finally identified in 2014 [99,100]. Up to five different isoforms (LRRC8A/B/C/D/E) have been found to form this channel (LRRC8-VRAC). LRRC8A (8A) is the essential component allowing the membrane trafficking of the heteromers, but leads to fully functional channels only in the presence of at least one of the other isoforms [99,100,101]. The molecular composition of the heteromers determines their biophysical properties, and thus inactivation kinetics [99,101,102], rectification and single-channel conductance [101] vary among the different combinations. Several reports have confirmed that the LRRC8-VRAC is able to mediate the transport of organic osmolytes, with permeability also being markedly affected by the subunit composition [99,101,102,103,104,105]. Thus, while the LRRC8D (8D) and LRRC8E (8E) subunits conferred similar d-aspartate (glutamate analogue) efflux rates when co-expressed with 8A, 8D showed a much broader spectrum of substrate permeability including GABA, taurine, myo-inositol and d-lysine. Conversely, LRRC8B (8B) and LRRC8C (8C) could both have a mainly modulatory effect on substrate specificity in heteromers containing more than two different isoforms [103]. Whether the VRAC is able to mediate ATP release is a controversial matter since ATP also inhibits the channel [97,98,106]. In *Xenopus* oocytes, ATP efflux has been detected in LRRC8 heteromers, especially through the 8A/8E combination [102]. 

### 4.2. Structure and Function

LRRC8 proteins share a certain sequence homology with pannexins, and a hexameric conformation has been proposed [107]. Very recently, four different research groups have obtained cryo-electron microscopy (CEM) structures of the 8A subunit, thereby providing the first insights into the channel structure [108,109,110,111]. Interestingly, it has been shown that the 8A subunit alone displays poor but still functional VRAC activity (8A-VRAC) when overexpressed in LRRC8^−/−^ cell lines, thus allowing for structure–function studies. 

The 8A-VRAC is a hexamer in which each subunit contributes to pore formation (Figure 3A). Interestingly, the 8A-VRAC shows structural homology with the gap junction channel connexins [112] and innexins [113], explaining to a certain extent why these proteins share cross-inhibition to known pharmacological compounds [114,115,116,117]. Both the N- and C-terminus in a single subunit are intracellular. A single 8A subunit has an extracellular domain (ED) with two extracellular loops, a transmembrane domain (TD) with four transmembrane helices, an intracellular linker domain (ILD), and the intracellular LRR domain (LRRD) with up to 15–16 leucine repeats (Figure 3A). A particular feature of the hexamer is the symmetry divergence between the LRR domains and the rest of the channel. While ED, TD and ILD show a six-fold symmetry axis, the LRRD assembles into a trimer of dimers conformation (Figure 3B). Several constrictions can be identified along the channel pore. The narrowest point is formed by a ring of arginines (R103) located in the N-terminal extreme of an α-helix (named the pore helix) in the ED. Mutations in R103 and the adjacent residue L105 altered the selectivity of the VRAC in co-expression with 8C, indicating that this highly electropositive region configures the selectivity filter of the channel [110,111]. Furthermore, the 8A R103F mutant suppressed the ATP blocking effect. Interestingly, a recent CEM report has shown that DCPIB also plugs into this constriction [109], thus suggesting the involvement/proximity of the region to the binding sites of different inhibitors. Adjacent to the R103 residue, mutations of C-terminal residues in the first extracellular loop (K98 and D100 (KYD motif—Figure 3C)) were previously shown to alter the selectivity and inactivation kinetics of the channel [118]. Indicating its flexibility, none of the CEM structures published so far has fully resolved the first extracellular loop (Figure 3C). Nevertheless, it seems to extend towards the neighbouring subunit, suggesting that the above-mentioned channel properties are influenced by conformational changes and interactions in this region. At the extracellular membrane boundary, mutations of the T44 residue were shown to significantly alter the Cl^−^ vs. I^−^ permeability, thereby suggesting it to be a pore-facing residue [100,101]. CEM structures have confirmed that T44 and T48 residues configure another important pore constriction (Figure 3C). At the intracellular side of the pore lies the unresolved N-terminus of the channel. Interestingly, mutations and/or thiol-reagent cysteine modifications of several residues in the region drastically affected channel function, including alterations in permeability as well as voltage-dependence [111,119]. Given the revealed structural homology with gap junction proteins, the N-terminus of the VRAC is also proposed to form a tight pore constriction that could couple LRRD’s conformational changes to the pore (Figure 3C) [112,113]. The big structural difference between gap junction proteins or pannexins and the VRAC relies on the C-terminus, which is extremely long in the VRAC. In all the 8A-VRAC CEM structures, the intracellular LRRDs (Figure 3A–C) show a poorly defined structure, suggesting flexibility and a possible role in channel gating. Both compact/constricted and relaxed/expanded conformations of the C-terminus have been found in the 8A-VRAC [108,109]. Taking into account the C-terminus of LRRC8A, the biophysical basis of ionic strength sensing has also been suggested. Thus, electrostatic interactions occurring between LRRDs could be affected by intracellular ionic strength, triggering conformational changes that could be coupled to the opening of other regions in the channel pore. The predicted phosphorylation sites in the LRRDs suggest that these conformational changes could also be modulated by intracellular kinases [120]. Recently, a heterozygous variant (R545H) identified in a SCOS (Sertoli cell-only syndrome) patient was found to significantly reduce VRAC activity [121]. In the LRRC8A structure (Figure 3C), R545 was one of the charged residues forming electrostatic interactions between adjacent LRR domains. In the same line, a previous observation by our group suggested the role of the LRRDs in channel gating: C-terminal fusion of the monomeric fluorescent proteins VFP and mCherry produced a shift in osmolarity-dependence, increased conductance, and the stable constitutive currents under isotonic conditions [102]. These results prompted us to propose a foot-in-the-door mechanism whereby the LRRDs conformation could be altered by the presence of the fluorescent proteins.

As CEM structural studies have mainly focused on the 8A-VRAC protein, we do not know how representative these structures are of heteromeric VRAC assemblies. Several pieces of evidence suggest that the basic structural properties described for the 8A-VRAC could be recapitulated in LRRC8-VRAC heteromers: (i) the lower-resolution CEM structure of 8A/8C heteromers has very similar structural features to the homo-hexamer [110]; (ii) Different purified populations of VRAC heteromers are the same size as 8A-VRAC on native gels, suggesting an identical oligomeric state [101]; (iii) Although less functional, the 8A-VRAC retains similar biophysical properties to the heteromers, as it is also inhibited by DCPIB; (iv) There is high sequence homology among the different LRRC8 isoforms, and virtual modelling suggests a high degree of structural conservation.

### 4.3. VRAC in Astrocytes

Astrocytes show a peculiar predisposition to swell under physiological [63,120,122], and pathological conditions (see Reference [120]). Due to its potential role in CNS physiology, the VRAC is of major interest in the astroglial context. Recent studies in cultured astrocytes have found that LRRC8 proteins are indispensable for the swelling-induced and ATP-induced release of neuroactive molecules like taurine, glutamate, aspartate, and myo-inositol, suggesting that different combinations of LRRC8 isoforms could form independent pathways for charged and uncharged molecules [104,105]. Furthermore, VRAC currents were also reduced after LRRC8A depletion [123,124]. Given that the molecular correlates of the VRAC have only recently been discovered, and given the poor pharmacological tools available alongside the absence of specific inhibitors of VRAC activity, the physiological and pathophysiological contribution of the astrocytic VRAC in the brain remains poorly understood. 

Astrocytes swell in response to several pathological conditions like trauma, strokes hyponatremia, and epilepsy [120]. In such conditions, it is thought that VRAC activation could lead to excitatory amino acid (EAA) release and excitotoxic neuronal damage. It has been proposed that under ischaemia, hypoxic conditions trigger astrocyte swelling and glutamate release through the VRAC. Among several glutamate receptors, glutamate is expected to activate Ca^2+^-permeable glutamate receptors like NMDA in neurons, leading to membrane depolarization. Prolonged glutamate build-up in the ECF would exert sustained neuronal depolarization, increased intracellular Ca^2+^ levels and finally apoptosis (Figure 3D). Among associated pathological conditions, the possible role of the VRAC in stroke has been most extensively studied in animal models, where the administration of the VRAC inhibitors tamoxifen [125] and DCPIB [126] was shown to markedly diminish brain damage (Figure 3D). It was also shown that VRAC inhibitors effectively reduced glutamate levels in the ischaemic region. However, given the poor specificity of the inhibitors, cross-inhibition opens the possibility that other glutamate pathways like connexins or the glutamate transporter (GLT-1) could be responsible for the glutamate build up, and the conclusion that VRAC activity is the primary cause has met some criticism [114,120]. Experiments in inducible cell-specific LRRC8 KO animals and/or the development of new specific inhibitors of the channel could be more conclusive.

Apart from its proposed roles in human pathologies, the astrocytic VRAC has been implicated in several physiological processes. Given its critical role in cell volume regulation, it is fair to assume a key contribution of the channel to different astroglial cellular processes, such as cell cycle or migration [96]. Besides this, other physiological roles could include extracellular signalling and K^+^-siphoning. Another proposed physiological role of the VRAC occurs in the supraoptic nucleus, where VRAC activation in astrocytes could modulate antidiuretic hormone release by neurons, thus controlling whole-body fluid homeostasis. These astrocytes are thought to swell upon systemic hypo-osmolarity and release large amounts of taurine through VRAC activation. In this model, taurine would activate glycine receptors in the surrounding neurons, triggering their hyperpolarization and the inhibition of vasopressin and oxytocin release [96,120]. Again, further studies using either animal models or improved pharmacological tools will be needed to strengthen the notion of the VRAC as an extracellular signalling modulator. 

As mentioned in the ClC-2 channel section, K^+^-siphoning requires associated Cl^−^ fluxes to guarantee the electroneutrality of the process, and MLC1-GlialCAM-ClC-2 have been found to form a ternary complex that could be important for this purpose [53]. Interestingly, *Mlc1* KO and knockdown astrocytes (and also GlialCAM knockdown) show reduced VRAC activity [124,127,128]. These studies suggested that the functional link between MLC1 and the VRAC could happen in an indirect manner involving signalling cascades. Although the possible contribution of the VRAC to the process remains unknown, it is expected that the net influx of KCl during K^+^-siphoning could trigger astrocyte swelling through alterations in osmotic gradients, raising the possibility that VRAC activity plays a role in controlling neuronal excitability. 

## 5. Maxi Cl^−^ Channels (MAC) 

### 5.1. General Features

The Maxi-Cl^−^ channel (MAC) is a ubiquitously expressed anion channel with particular biophysical properties. The most prominent characteristic is its large single-channel conductance of 300–500 pS, making it a highly efficient anion-transporting pathway [129]. Other “fingerprint” properties include the following [129]: (i) an ohmic current–voltage (I–V) relationship; (ii) maximal open probability (P_open_) at around 0 mV but inactive when the membrane potential changes by as much as about ±20 mV; (iii) strong selectivity preference for anions over cations, with PCl/PNa > 8; and (iv) sensitivity to extracellular Gd^3+^ ions. Interestingly, the channel is permeable to large molecules like pyruvate, glutamate, and even ATP [130]. 

MACs are normally inactive under resting conditions and can be activated by several stimuli, such as osmotic swelling, hypoxia, salt stress or patch excision. Cytoskeleton and signalling pathways mediated by GPCRs and intracellular-calcium-level-modifying agents have been proposed to modulate channel activity, perhaps at least in part by altering the phosphorylation state through intracellular kinases or phosphatases [129].

Very recently, SLCO2A1 was identified as the core component of the MAC in C127 cells [131]. Surprisingly, SLCO2A1 was previously described as a prostaglandin transporter (PGT) [132]. Therefore, the protein seems to have a dual role as a transporter and a channel; the authors also confirmed that the SLCO2A1-MAC is permeable to ATP.

### 5.2. Structure–Function

Given the absence of crystal structures for the SLCO2A1 channel/transport protein, the known structure–function features are entirely based on mutagenesis and indirect modelling approaches. The SLCO2A1 gene (NP_005621.2) encodes a 643 amino acid polypeptide with up to 12 predicted transmembrane segments. Having permeability for ATP, which has a mean radius of ∼0.58–0.65 nm [133], indicates that the MAC should have a larger pore radius. In fact, estimations propose a medial constriction of ∼0.55–0.75 nm, with an extracellular radius of ∼1.42 nm, and an intracellular radius of ∼1.16 nm [129]. 

Several residues in the C-terminal side of the putative transmembrane 10 were proposed to be part of the PGE2 binding site [134]. Later studies allowed for the identification of two conserved positively charged residues (R560 (α-helix 11) and K613 (α-helix 12)) involved in substrate translocation and binding. The charge-neutralized mutant R560N was non-functional, while K613G was still functional, but exhibited a 93% reduction in transport rate [135]. Interestingly, recent studies showed that the mutants have reduced single-channel current amplitude, altered gating kinetics and voltage dependence, especially at positive voltages. Furthermore, K613G altered the selectivity of the channel by increasing cation permeability, suggesting that it could be a pore-facing residue. Surprisingly, the K613G mutation had no apparent effect on swelling-induced ATP release by the heterologous expression in HEK293T cells, showing similar values to the wild-type (WT) protein [131]. In addition, the disease mutants G222R and P219L (previously associated with pachydermoperiostosis [136,137]) were also found to be non-functional [131]. By homology modelling with the related glycerol-3-phosphate transporter, it was proposed that the R560 residue could be located in the central axis of the protein, suggesting that this location could be important for determining voltage-dependent inactivation and channel function. Consistent with this assumption, non-functional disease mutants P219 and G222 are also located close to the central axis (Figure 4A,B). However, the K613 location was farther away from the central axis (Figure 4A,B). Given the impact of the mutation on the biophysical properties of the channel, and the fact that homology modelling was based on a related transporter structure, it was hypothesized that the model could represent an “inward-open transporter conformation”. Thus, it was suggested that the SLCO2A1 protein could have two different conformations: a resting PGT transporter conformation and an activated MAC conformation (Figure 4C). According to this hypothesis, in the activated conformation the K613 residue could come closer to the central axis and contribute to the selectivity filter [131].

Nevertheless, given the absence of MAC channel crystal structures, many questions remain to be resolved. Although an oligomeric assembly of the channel has previously been suggested [129], it remains unknown whether the channel/transporter pore is formed by a single subunit or by multiple subunits. Another unknown question is the possibility that different proteins or auxiliary subunits could join the SLCO2A1 protein as part of a more complex assembly of the MAC. Several pieces of evidence support this notion. First, a non-Maxi-Cl^−^ type ion channel was identified in SLCO2A1-deficient C127 cells. Interestingly, this non-Maxi channel was sensitive to the MAC inhibitor bromosulfophthalein (BSP), while the purified and reconstituted SLCO2A1-MAC was insensitive. These results suggested that an orphan auxiliary subunit of the SLCO2A1 protein could be responsible for the non-Maxi-Cl activity in association with a different membrane protein, and perhaps could also be the BSP-sensitive partner of the MAC complex [131]. Second, the fact that MAC in membrane blebs and reconstituted SLCO2A1-MAC in proteoliposomes were constitutively activated is also compatible with the absence of auxiliary/modulatory components of the MAC channel. Third, the voltage-dependent inactivation of the endogenous MAC in C127 cells was markedly stronger than that exhibited by the heterologously expressed SLCO2A1-MAC in HEK293T cells (without endogenous MAC activity), suggesting differences in endogenous modulatory components. Further experiments are needed, but in vivo co-immunoprecipitation experiments, as well as proteomics assays, could be interesting in order to search for the possible interacting proteins in the putative MAC complex. 

### 5.3. The MAC in Astrocytes

Osmotic swelling [138,139], chemical ischaemia and hypoxia [139,140] are stimuli previously reported to activate the MAC in astrocytes. Thus, it has been suggested that the astrocytic MAC makes an important contribution to brain physiology and pathophysiology [129]. However, the recent discovery that the SLCO2A1 prostaglandin transporter (PGT) is also the core component of the MAC enables a critical review of these functions. Previous studies in human patients and mouse models have revealed the physiological impact of the SLCO2A1 deficiency. According to “The Human Gene Mutation Database”, up to 41 mutations have been identified in patients. Two of these mutations have recently been found to impair channel function [131], and several are also expected to abolish either channel and/or transport activity. Perhaps the most extreme case is a homozygous patient harbouring an early nonsense mutation (G104X) in the SLCO2A1 gene, expected to codify a non-functional protein, given that it eliminates the majority of the polypeptide [141]. As usually happens in patients with mutations in the *SLCO2A1* gene, this patient suffered from pachydermoperiostosis (PDP), but no cognitive/neurological clinical manifestations were identified [142]. Furthermore, *slco2a1* deficiency in mouse models caused the early postnatal death of the animals, presumably because PGE_2_ accumulation impairs ductus arteriosus closure. This defect could be rescued by the administration of indomethacin before birth, but the mice showed no apparent abnormalities other than increased excretion of PGE_2_ and lower plasma levels [143]. Expression studies have identified lower expression of PGT in the brain than in other tissues [132,144]. Although more experiments should be performed, this evidence strongly suggests that the SLCO2A1 protein is not crucial in either astrocytes or brain physiology. 

Different possibilities could explain the above evidence: (1) the SLCO2A1 protein is not the core component in the astrocytic MAC, and another similar protein could replace it as part of a different MAC protein complex with similar biophysical properties; (2) although the SLCO2A1 protein is physiologically important in the brain, other channels/transporters, perhaps upregulated, are able to perfectly compensate for its deficiency; (3) MAC activity is not relevant under physiological conditions in the brain. 

At the pathophysiological level, early controversial results can be found in the literature about the main channel responsible for ischaemia swelling-induced glutamate release in astrocytes. Some authors suggested that ischaemia swelling-induced glutamate release in cultured astrocytes could be mediated simultaneously by both the VRAC and the MAC [139]. However, other studies suggested that the VRAC is mainly responsible [145]. As discussed in the VRAC section, in vivo studies in animal models of stroke confirmed that VRAC inhibitors like DCPIB and tamoxifen effectively reduced glutamate build-up and cerebral damage. Later studies in cultured astrocytes also suggested that glutamate build-up is a DCPIB-sensitive process [114]. Given that MAC are insensitive to DCPIB [129], these results strongly suggested a preferential role of the VRAC and/or other DCPIB-sensitive glutamate pathways in ischaemia. Recent studies in cultured rat astrocytes have confirmed the expression of LRRC8 isoforms, indicating that swelling- and the ATP-induced release of neuroactive molecules, including that of glutamate, is almost exclusively dependent on LRRC8A expression, suggesting that different combinations of LRRC8 isoforms could form independent pathways for charged and uncharged molecules [104,105]. The abovementioned studies also support the small contribution of the MAC to the pathological glutamate release under ischaemic conditions. Nevertheless, further experiments are needed. Despite *slc02a1* identification, no author has yet confirmed the expression of the protein in astrocytes, and reliable experiments to study its possible role in ischaemic glutamate release remain to be performed. 

Various stimuli have been shown to trigger ATP release from astrocytes [146,147,148,149,150,151,152] (Figure 4C). However, it remains a matter of discussion whether there is a main pathway and which ion channels are responsible for this release. Besides exocytotic ATP release, several ion channels have been suggested to mediate ATP release in astrocytes, including connexin-43 [153,154], pannexin-1 [155,156], and MAC [140,157]. As VRAC has been found to mediate ATP release in endothelial cells [97], they may also be important in astrocytes. Some of these channels show overlapping pharmacology (i.e., Cx43, Pannexin-1, VRAC), which enormously complicates the interpretation of previous results. New astrocyte-specific KO mice will be necessary to unravel the contribution of each channel to the process of ATP release.

## 6. Summary and Outlook 

In the last decade, the number of studies on astrocytic chloride channels has grown tremendously, moving from the functional characterization of the chloride channel activities present in astrocytes (for instance, the VRAC) to identifying the molecular nature (LRRC8) and even 3D structures of the proteins involved in these activities. In some cases, like the ClC-2 chloride channel, the identification of regulatory proteins (GlialCAM) has provided insights into the physiological role of the channel. We believe that the next few years will provide new mechanistic details about how these proteins work, through a combination of structural biology, mutagenesis and functional measurements. The generation of conditional astrocyte-specific knockout mice will be very important to determine the contribution of each chloride channel to brain physiology. The identification of missing subunits and regulatory proteins could also be crucial to understanding their physiological role. We believe that many new surprises are waiting to be discovered.

## Figures and Tables

**Figure 1 ijms-20-01034-f001:**
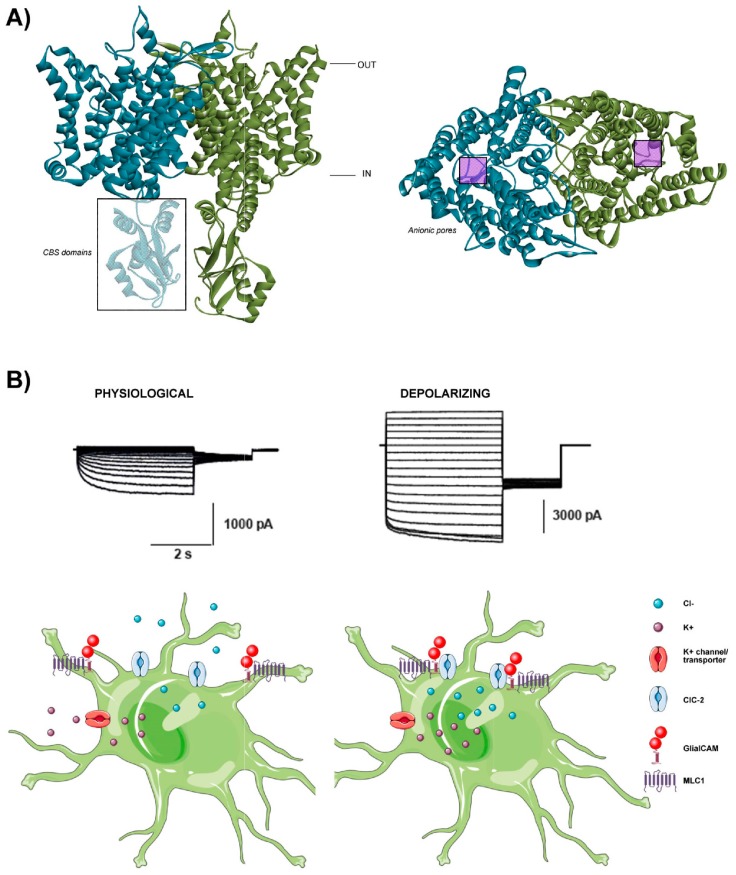
The chloride channel ClC-2. (**A**) The structural configuration of a CLC chloride channel. (Left) Lateral view of the dimeric form of the channel, which is comprised of two subunits distinguished by different colours. The intracellular cystathionine-β-synthase (CBS) domains are also highlighted in the unshaded box. (Right) Frontal view of the dimeric channel, where the two pores are highlighted in the purple boxes. Figures generated using the PDB model 5TQQ from human ClC-1, which has high homology to the ClC-2 channel. (**B**) Proposed model for depolarizing-mediated ClC-2 activation by MLC1/GlialCAM in astrocytes. In basal conditions, ClC-2 displays outwardly rectifying currents, promoting Cl^−^ efflux (left). Upon depolarization and the consequent entrance of K^+^ into the cell, the MLC1/GlialCAM complex binds to ClC-2, allowing GlialCAM to modify ClC-2-mediated currents, generating a Cl^−^ influx into the cell that may compensate the excess of K^+^ (right).

**Figure 2 ijms-20-01034-f002:**
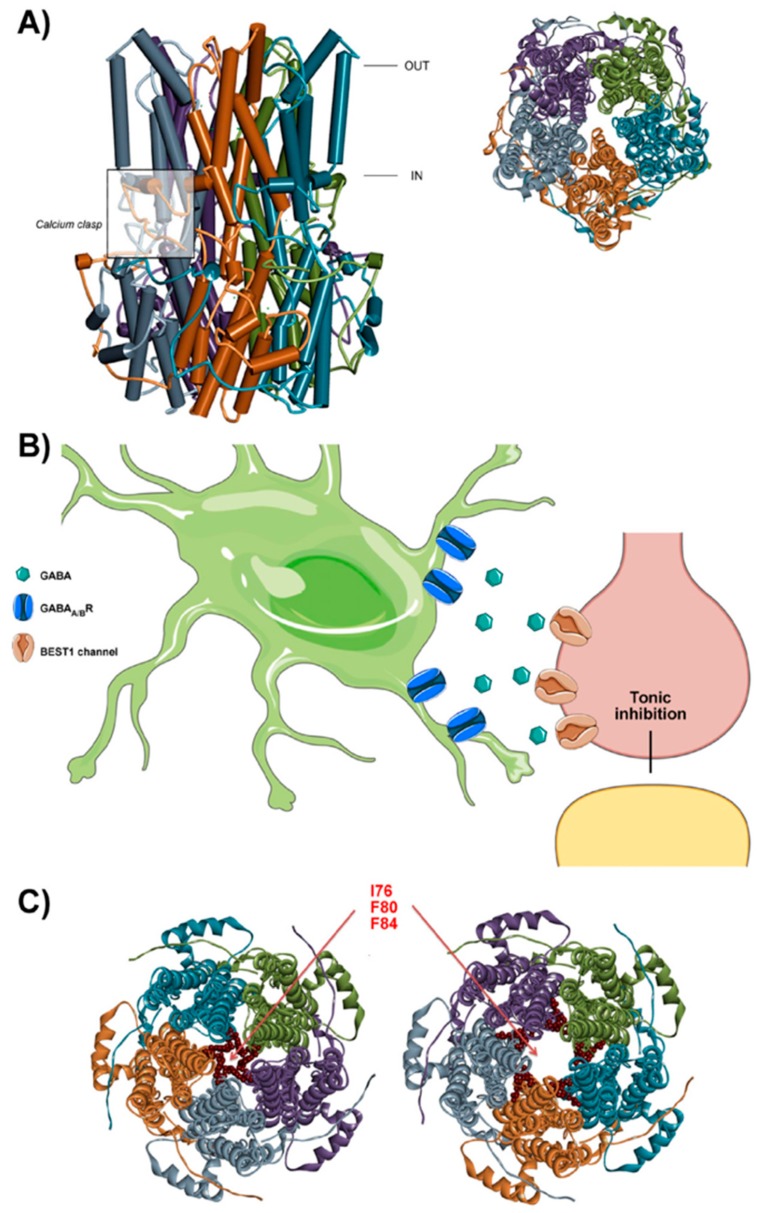
The bestrophin chloride channel. (**A**) The structural configuration of the Best1 channel. (Left) The schematic topology of Best1 channel, with the transmembrane and intracellular domains clearly identified and the calcium clasp highlighted in the box. (Right) Frontal view of the pentameric channel, where the five subunits are differentiated by colour and the central pore is visible in the centre of the channel. Figures generated using the PDB model 4RDQ from chicken Best1 protein. (**B**) Schematic model of Best1-mediated tonic inhibition in the cerebellum, where astrocytes release GABA through Best1, which acts on neuronal GABAergic receptors, which in turn may be relevant for motor coordination. (**C**) Closed (left) and open (right) configurations of the Best1 channel, with the residues conforming the hydrophobic seal (I76, F80, F84) highlighted in red. Note the significant change in the size of the pore in the open (Ca^2+^ bound) configuration. Figures were generated using the PDB models 6N23 and 6N28 from the structure of the chicken Best1 channel.

**Figure 3 ijms-20-01034-f003:**
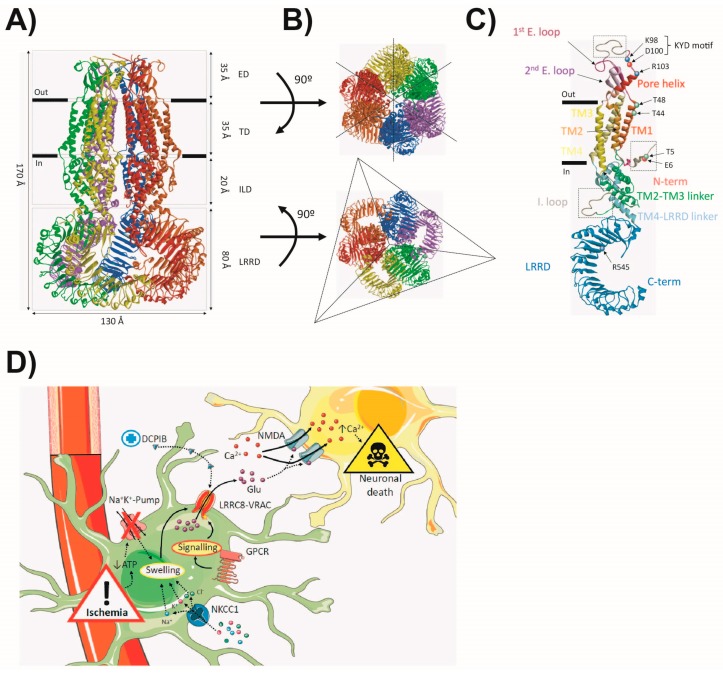
The VRAC. (**A**) The cryo-based model of the 8A-VRAC viewed from the membrane plane. The different subunits of the hexamer are highlighted in different colours, and domain layers are indicated by dashed rectangles. (**B**) Extracellular (top) and intracellular (bottom) views of the 8A-VRAC model. Dashed lines highlight the symmetry axis of the structures, with a six-fold symmetry axis observed in the top view, and a three-fold symmetry axis in the bottom view (LRRD). (**C**) Representation of a single LRRC8A subunit. Different helices and domains are indicated in different colours. Dashed squares delimit unresolved regions in the structure. Relevant positive residues (blue), negative (red) and polar (green) are indicated with coloured spheres. (**D**) The hypothetical model of the pathophysiological role of the astrocytic VRAC in ischaemia. Ischaemic conditions lead to depletion of intracellular ATP due to unsustainable oxidative metabolism, triggering the inactivation of the Na^+^/K^+^-pump, and subsequently cell swelling. NKCC1 cotransporter activity could further contribute to cell swelling. Astrocyte swelling triggers the opening of the LRRC8-VRAC mediating glutamate efflux. Glutamate build-up in the ischaemic penumbra produces its excitotoxic effect through the overactivation of glutamate-gated channels like NMDA, leading to sustained depolarization and increased Ca^2+^ levels, thus triggering neuronal damage and death. GPCR signalling could also modulate LRRC8-VRAC activation, but the possible contribution to the process is yet to be discovered. Pharmacological compounds like DCPIB could exert their protective effect by blocking the LRRC8-VRAC, thus reducing glutamate build-up in the ischaemic penumbra.

**Figure 4 ijms-20-01034-f004:**
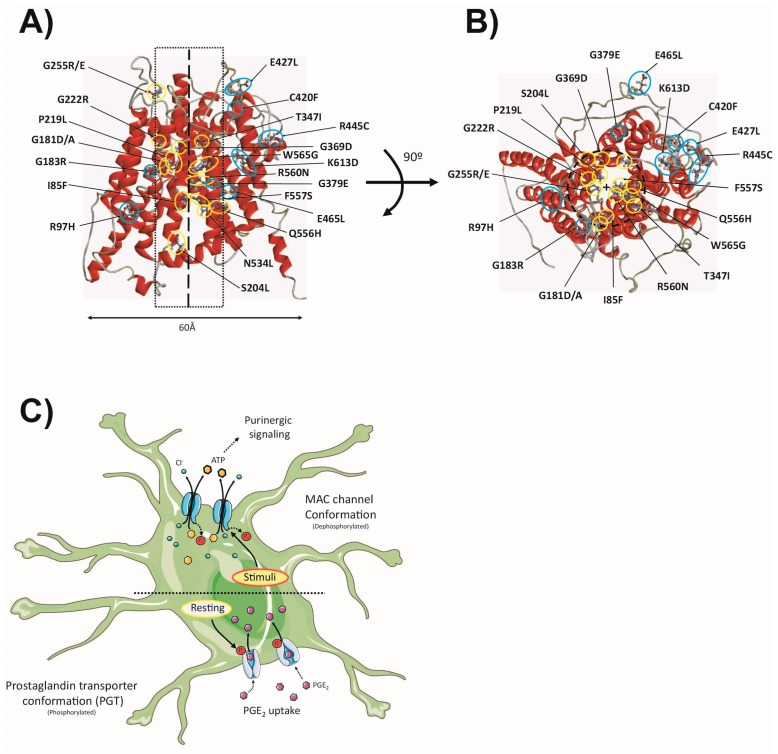
The SLCO2A1 protein, homology model and physiological role. (**A**) A 3D homology model of the SLCO2A1 protein based on the crystal structure of the *E. coli* glycerol-3-phosphate transporter (PDB ID: 1PW4A), constructed using the I-TASSER® software, as described elsewhere [139]. The protein is shown from the membrane plane. All the missense mutations found in pachydermoperiostosis patients are shown, including the charge-neutralized mutants, K613G and R560N. The thick dashed line indicates the central axis of the protein, and the thin dashed rectangle delimits a proximity threshold to the central axis. The residues that are close to (yellow), or far from (blue) the central axis, are highlighted with coloured circles. Notice the preferential location of most mutants to the vicinity of the central axis. (**B**) The top view of the homology model. The central cross represents the central axis, and the dashed circle delimits the proximity area. (**C**) The hypothetical physiological role of the SLCO2A1 protein. The horizontal dashed line in the centre of the cell highlights the dual behaviour of the protein either in the prostaglandin transporter (PGT) conformation (down) or the Maxi-Cl^−^ channel (MAC) conformation (up). Under resting conditions, the phosphorylated SLCO2A1 protein behaves as the PGT, allowing for PGE_2_ uptake and its clearance from the extracellular fluid. Various stimuli (osmotic swelling, hypoxia, salt stress, patch excision, etc.) can trigger protein dephosphorylation, and the activation of the SLCO2A1-MAC activity (up). Released ATP would be able to mediate autocrine or paracrine purinergic signalling.

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
