# Peer review of "Chloride Channels in Astrocytes: Structure, Roles in Brain Homeostasis and Implications in Disease"

_ijms, 2019, doi:10.3390/ijms20051034_

Round 1
Reviewer 1 Report
This is a comprehensive and thorough review of astrocyte chloride channels. It will form a basis for moving forward with studies of astrocyte chloride channel function in the future.
Author Response
We thank the reviewer for his/her positive comments.
Reviewer 2 Report
This review summarizes molecular properties and purported functional roles of chloride channels in astrocytes, the most abundant and the most functionally diverse cell type in the brain. In contrast to the well-appreciated roles for the neuronal ligand-gated GABA and Gly receptor-channels, the significance of chloride channels in glial cells has only began to be recognized. Nevertheless, recent developments suggest the important role of astrocytic chloride signaling in cell autonomous and paracellular regulation of broader brain functions. For this reason, the present manuscript is a welcome addition to the field.
What positively discriminates the present work from other recent publications on the same topic, is the comprehensive nature of the manuscript and unusually great insight into molecular structure of very diverse chloride channel classes, which are rarely considered together. I have no problem with the scientific content of this manuscript, with one caveat.
Specific comments:
1. The overview of the functional significance of Bestrophin-1 channels is based on publications from just one group of C. Justin Lee, which are not without controversy. Transcriptomic analysis (see for example data from Ben Barres’s RNAseq database) suggest very low astrocytic expression of Best-1, inconsistent with many major functions that have been assigned to this chloride channel. Furthermore, contributions of Best-1 to tonic GABA release has been questioned because blockers of volume-regulated and Best 1 anion channels have off-target non-specific effects on GABA-A receptors (e.g., see discussion in K. Le Meur et al., Front. Comp Neurosci. 2012). To the best of my knowledge, the missense mutations of Best1, which produce Best1 vitelliform macular dystrophy are not associated with any distinct brain phenotypes. Perhaps, these considerations deserve a disclaimer.
2. The moderate, non-scientific issue with this manuscript is related to suboptimal English. Although the language is quite comprehensible, the manuscript contains numerous grammatical errors. These are too many too list them. I recommend to the Authors recruiting language editing services. Their review is too good and certainly deserves this extra effort and expense.
3. Another moderate problem is erroneous citation style. Numerous citations in the list of references do not include proper year, page numbers, and/or name of the journal. Please correct references [3, 7, 12, 24, 44, 54, 55, 119, 125, 130, 136, 144, 150, 152, 162]. Perhaps, there are more.
4. Minor point, On p. 1, ln. 30, the Authors refer to “active Cl- transporters”. This is incorrect, as NKCCs and KCCs are SECONDARY-active Cl- transporters, driven by electrochemical ionic gradients.
5. Representative Cl- traces in Fig. 1B are too small to make a meaningful point. I think that they deserve an extended space.
6. In Fig. 3C it would be helpful to show membrane boundaries.
7. In Fig. 4C, refer to the MAC channel conformation of SLCO2A1 and the prostaglandin transporter conformation of SLCO2A1.
8. P. 14, ln. 548: identifying the molecular nature (or relevant genes but not names).
Author Response
We thank the reviewer for his/her positive comments about our review.
Specific comments
Functional significance of Bestrophin-1 channels.
We agree with the reviewer that all the articles about the physiological role of Bestrophin come from the group of C. Justin Lee. The same type of criticism could be applied to the work with the MAC. Taking into account the specific comments raised by the reviewer: 1) mRNA levels are low in astrocytes: The work done by Justin Lee's group to detect Best1 expression in Bergmann glia, for instance, is based on Best1 antibodies using knockout mice as a negative control. 2) Off-target effects of non-specific blockers: We agree with this criticism. However, recently the same group has done these experiments using Best Knockout mice. Therefore, based on these new results, the functional role of Best1 in astrocytes is more convincing. However, we add a paragraph in the review mentioning Ben Barres's RNAseq database.
Suboptimal English: We were surprised about this comment since we send or manuscript before submitting to a language editing service, which is very expensive. The same editing service has reviewed again our manuscript incorporating new changes. Furthermore, we have looked carefully at the whole manuscript again. All the changes are tracked in green.
We are sorry for this mistake, which is due to a problem with the new version of the bibliographic software (Mendeley). We have corrected all the reference manually.
Secondary transporters. Right. It has been corrected.
Traces in Figure 1B. We have changed the corresponding figure.
Membrane boundaries in Fig. 3C. Done.
Figure 4C. The legend of the figure has been changed.
Molecular nature: Done.
We thank the reviewer for his/her comments that improved our review.